# Post-Electric Current Treatment Approaching High-Performance Flexible n-Type Bi_2_Te_3_ Thin Films

**DOI:** 10.3390/mi13091544

**Published:** 2022-09-17

**Authors:** Dongwei Ao, Wei-Di Liu, Fan Ma, Wenke Bao, Yuexing Chen

**Affiliations:** 1School of Machinery and Automation, Weifang University, Weifang 261061, China; 2Australian Institute for Bioengineering and Nanotechnology, The University of Queensland, St Lucia, Brisbane, QLD 4072, Australia; 3School of Materials Science and Engineering, Inner Mongolia University of Technology, Hohhot 010051, China; 4Shenzhen Key Laboratory of Advanced Thin Films and Applications, Key Laboratory of Optoelectronic Devices and Systems of Ministry of Education and Guangdong Province, College of Physics and Optoelectronic Engineering, Shenzhen University, Shenzhen 518060, China

**Keywords:** thermoelectric, Bi_2_Te_3_, flexible, thin film, electric current treatment

## Abstract

Inorganic n-type Bi_2_Te_3_ flexible thin film, as a promising near-room temperature thermoelectric material, has attracted extensive research interest and application potentials. In this work, to further improve the thermoelectric performance of flexible Bi_2_Te_3_ thin films, a post-electric current treatment is employed. It is found that increasing the electric current leads to increased carrier concentration and electric conductivity from 1874 S cm^−1^ to 2240 S cm^−1^. Consequently, a high power factor of ~10.70 μW cm^−1^ K^−2^ at room temperature can be achieved in the Bi_2_Te_3_ flexible thin films treated by the electric current of 0.5 A, which is competitive among flexible n-type Bi_2_Te_3_ thin films. Besides, the small change of relative resistance <10% before and after bending test demonstrates excellent bending resistance of as-prepared flexible Bi_2_Te_3_ films. A flexible device composed of 4 n-type legs generates an open circuit voltage of ~7.96 mV and an output power of 24.78 nW at a temperature difference of ~35 K. Our study indicates that post-electric current treatment is an effective method in boosting the electrical performance of flexible Bi_2_Te_3_ thin films.

## 1. Introduction

Flexible thermoelectric (TE) devices, with the advantages of being self-powering, sustainable, and of small volume, provide a reliable power supply solution for wearable electronics, implantable electronics, and chip-sensors at near-room temperature [1,2,3]. The main challenge for flexible TE devices lies in the TE material performance and device integration technology. The performance of TE materials is evaluated by dimensionless figure-of-merit *ZT* (*ZT* = *S*^2^*σT*/*κ*, where *S*, *σ*, *S*^2^*σ*, *T*, and *κ* represent Seebeck coefficient, electrical conductivity, power factor, absolute temperature, and thermal conductivity, respectively) [4,5]. Recently, the flexible thin film (f-TF) provides an avenue for flexible TE devices due to the excellent flexibility, comparing with the bulk TE counterparts [6,7]. Organic f-TFs, including P3HT (*S*^2^*σ* < ~0.04 μW cm^−1^ K^−^^2^ at room temperature) [8], PEDOT:PSS (*S*^2^*σ* < ~0.5 μW cm^−1^ K^−2^ at 300 K) [9], and PANI (*S*^2^*σ* < ~0.6 μW cm^−1^ K^−2^ at 300 K) [10], are typically highly flexible with relatively low TE performance compared with the inorganic f-TFs. Inorganic f-TFs with excellent TE performance have received extensive attention, such as SnSe (*S*^2^*σ* = ~3.5 μW cm^−1^ K^−2^ at 300 K) [11], CuI (*S*^2^*σ* = ~3.75 μW cm^−1^ K^−2^ at 300 K) [12], Ca_0.35_CoO_2_ (*S*^2^*σ* = ~0.5 μW cm^−1^ K^−2^ at 300 K) [13], Ag_2_Se (*S*^2^*σ* = ~9.874 μW cm^−1^ K^−2^ at 300 K) [14], and Bi_2_Te_3_-based f-TFs (*S*^2^*σ* = ~25 μW cm^−1^ K^−2^ at 300 K) [15].

Among many inorganic f-TFs, Bi_2_Te_3_ based ones are the most widely applied due to the excellent TE performance at room temperature [16,17]. Wu et al. [18] reported that hybridizing Bi_2_Te_3_ f-TFs with graphene oxide by vacuum filtration and annealing, and an *S*^2^*σ* of ~1.08 μW cm^−1^ K^−2^, is approached at ~297 K. Chen et al. [19] successful prepared Bi_2_Te_3_ nanowire-based f-TFs with an *S*^2^*σ* o of 1.10 μW cm^−1^ K^−2^ at 400 K by solution phase printing methods. Madan et al. [20] successfully fabricated Se-doped Bi_2_Te_3_ based f-TFs with the *S*^2^*σ* of ~2.65 μW cm^−1^ K^−2^ at ~297 K by mechanically alloyed and dispense printing method. Bi_2_Te_3_ f-TFs fabricated by in situ solution method has approached ~7.4 μW cm^−1^ K^−2^ at ~297 K [21]. Bi_2_Te_3_ f-TFs fabricated by thermal diffusion methods can achieve the *S*^2^*σ* of ~14.65 μW cm^−1^ K^−2^ at room temperature [22]. Additionally, many post-treatment techniques have been used to further improve the TE performance of n-type Bi_2_Te_3_ based f-TFs, such as such as heat treatment [23], laser treatment [24,25], infrared treatment [26], and electric current treatment [27].

Electric current treatment, as an effective and fast method, has attracted research interest [28]. Tan et al. [29] strengthened the anisotropy of electron mobility of Bi_2_Te_3_ based thin films by introducing electric current during the deposition process, and achieved a high *S*^2^*σ* of 45 μW cm^−1^ K^−2^. Zhu et al. [27] also used post-electric current treatment (P-ECT) methods to optimize phase transformations and crystal orientation of Bi_0.5_Sb_1.5_Te_3_ thin film, resulting in an increase in *σ*. It is typically understood that P-ECT can enhance the recrystallization kinetics, promote dislocation movement, and facilitate the formation of oriented microstructures in a short time [30,31]. It was worth mentioning that the thermal annealing effect is Joule thermal effect, and the athermal effect was mainly attributed to the electronic wind on atom diffusion [31]. Further research will analyze the effect of thermal effect and athermal effect on the doped Bi_2_Te_3_ f-TFs, respectively.

In this study, the magnetron sputtering is combined with P-ECT to prepare n-type Bi_2_Te_3_ f-TF on polyimide (PI) substrate (Figure 1a,b). Figure 1c shows an optical image of a typical n-type Bi_2_Te_3_ f-TF. Through optimizing the P-ECT current, the increase of carrier concentration (*n_e_*) is achieved, leading to a high *σ* of ~2065 S cm^−1^. The corresponding *S^2^σ* is ~10.70 μW cm^−1^ K^−2^, which is comparable with the reported n-type Bi_2_Te_3_ f-TF (Figure 1d). Applications of as-prepared n-type Bi_2_Te_3_ f-TFs were investigated via an assembled TE device, which is composed of 4 n-type Bi_2_Te_3_ legs. The device can generate the maximum open circuit voltage of ~7.96 mV and the maximum output power of 24.78 nW at the temperature difference (*∆T*) of ~35 K.

## 2. Experimental Section

The *n*-type Bi_2_Te_3_ f-TFs were deposited on a flexible PI substrate using a magnetron sputtering method. The deposition parameters of the thin film are presented as follows: the working pressure of 1 Pa, radio frequency sputtering power of 50 W, the sputtering temperature of 573 K, the background vacuum of 7.0 × 10^−4^ Pa, and argon flow of 40 Sccm. The KPS-3005D generator with the maximum output of 5.000 A was used to provide an electric pulse current. Bi_2_Te_3_ f-TFs were post-treated by electric current with duration of 1 s and interval of 1 s. The electric current was set as 0.3 A, 0.5 A, and 0.6 A, respectively, and the electric current time was 10 min. The thickness range of the Bi_2_Te_3_ f-TFs was ~580 nm. Finally, the flexible thermoelectric device was assembled with 4 n-type single-legs.

X-ray diffraction (XRD, D/max 2500 Rigaku Corporation, Tokyo, Japan, CuKα radiation) was employed to investigate the crystal structures of as-prepared Bi_2_Te_3_ f-TFs. SEM (Zeiss supra 55) was used to characterize the surface morphology. EDS (Bruker Quantax 200) was used to analyze the compositions of Bi_2_Te_3_ f-TFs. Hall measurement system (HL5500PC, Nano metrics) was employed to record *n_e_* and mobility (*μ*) values. A profilometer (Dektak XT, BRUKER, Germany) was employed to measure the thickness of flexible n-type Bi_2_Te_3_ thin films. And *σ* and *S* were simultaneously measured by the SBA458 (Nezsch, Germany).

## 3. Results and Discussion

XRD patterns were employed to analyze the crystal structure of as-prepared Bi_2_Te_3_ f-TFs as shown in Figure 2a. As can be seen, all the diffraction peaks can be indexed as the Bi_2_Te_3_ (PDF#15-0874), and no impurity peaks were observed. The right inset of Figure 2a shows the enlarged (006) peaks. The strongest peaks of all XRD patterns can be indexed as (006), indicating (*00l*) preferred orientation of all as-prepared Bi_2_Te_3_ f-TFs. Appendix A shows that no obvious crystallinity changes have been observed due to similar peak intensity and Full-Width Half-Maximum. Figure 2b shows a typical SEM-EDS pattern of Bi_2_Te_3_ f-TFs treated under the electric current of 0.5 A. The chemical compositions of as-prepared Bi_2_Te_3_ f-TFs are shown in Figure 2c and Table 1. Before P-ECT, the as-deposited Bi_2_Te_3_ f-TF presents the standard chemical stoichiometric ratio of ~2:3. With the increase of electric current, the Te content decreases due to the release of the unstable Te. With increasing the electric current from 0 to 0.6 A, the Bi content increases from 39.29 to 42.52 *at.* %, and Te content decreases from 60.71 to 57.48 *at.* %, indicating the increasing content of Te vacancies

To characterize the morphology of Bi_2_Te_3_ grains, the SEM images of as-prepared Bi_2_Te_3_ f-TFs treated under the electric current 0, 0.5, and 0.6 A are shown in Figure 3a–c, respectively. As can be seen, the as-prepared BT f-TFs are composed of hexagonal flakes stacking parallel to substrate. As the electric current increases from 0 A to 0.6 A, larger Bi_2_Te_3_ grains can be observed. Figure 3d shows the average grain size of as-prepared Bi_2_Te_3_ f-TF as a function of electric current. With the increasing of the electric current from 0 to 0.6 A, the average grain size increased from ~168 to ~381 nm. Appendix A compares the morphologies of as-prepared 0.6 A-Bi_2_Te_3_ thin film before and after cycling measurement of TE performance, where no obvious difference has been observed, indicating excellent stability. The grain growth with increasing electric current can be attributed to the additional energy supply from post-electric treatment [28,30].

Room temperature TE performance of as-prepared Bi_2_Te_3_ f-TFs is shown in Figure 4. Figure 4a shows the room temperature *σ*, *S*, and *S^2^σ* for Bi_2_Te_3_ f-TF as a function of electric current. The *σ* increases from 1874 to 2240 S cm^−1^ with increasing the electric current from 0 to 0.6 A, and the |*S*| decreases from 74 to 61 μV K^−1^. To better understand the change of *S* and *σ*, the *n_e_* was measured as shown in Figure 4b. The *n_e_* increases from 2.03 × 10^20^ to 3.84 × 10^20^ cm^−3^ with the increase of electric current. A simple relationship between *n_e_* and *S* can be exhibited by Mott formula [35]:(1)$$S=\frac{8\pi^{2}k_{B}^{2}T}{3{\mathrm{eh}}^{2}}m_{\mathrm{DOS}}^{*}{\left(\frac{\pi }{3n_{e}}\right)}^{2/3}$$
where *K_B_*, e, h, and m^*^_DOS_ present Boltzmann constant, electron, Planck Constant, and, the density of state’s effective mass, respectively. The reduced |*S*| is attributed to the increase of *n_e_* according to their inverse relationship between S and *n_e_* as expressed in Equation (1). Furthermore, according to the relationship between *σ* and *n_e_* as expressed in formula *σ* = *μ*e*n_e_*, the increase of *σ* is mainly attributed to the increase of *n_e_*. It is worth mentioning that the |*S*| of Bi_2_Te_3_ f-TFs is still lower than that of bulk materials due to the high *n_e_* > 1 × 10^20^ (detailed discussion in Appendix A). And the increased *n_e_* should be mainly attributed to the increased amount of Te vacancies with increasing the electrical current. The room temperature *S^2^σ* of Bi_2_Te_3_ f-TF as a function of electric current is shown in Figure 4a. The maximum *S^2^σ* of ~10.70 μW cm^−1^ K^−2^ can be achieved mainly due to the high *σ* of ~2065 S cm^−1^ and moderate *S* of −72 μV K^−1^. The TE performance tests of the 0.5 A-Bi_2_Te_3_ f-TF were repeated 3 times to verify the stability of as-prepared Bi_2_Te_3_ f-TFs as shown in Appendix A. Nearly unchanged TE performance during successive measurement cycles indicates high stability of our Bi_2_Te_3_ f-TFs. Element-doped Bi_2_Te_3_ based thin films usually have higher *S^2^σ* [36,37,38], and further research will analyze the effect of the electric current treatment on the doped Bi_2_Te_3_ f-TFs.

The bending tests were employed to investigate the bending resistance of as-prepared n-type Bi_2_Te_3_ f-TFs. Figure 5a,b shows the change of relative resistance (*∆R*/*R_0_*) as a function of bending cycles and bending radius, respectively. With the increase of cycles from 200 to 1000 under the bending radius of 9 mm, the *∆R*/*R*_0_ increases from 3.39% to 8.34% as shown in Figure 5a. In addition, with the increase of bending radius from 7 mm to 13 mm, the *∆R*/*R*_0_ decreases from 9.98% to 2.41% as shown in Figure 5b. The *∆R*/*R*_0_ < 10% suggests that the Bi_2_Te_3_ f-TFs possess excellent bending resistance [22,39]. Appendix A shows the repetitive test result of the bending resistance before and after cycling TE performance measurement, where high mechanical stability has been demonstrated. To demonstrate the practical applicability of Bi_2_Te_3_ f-TFs, a flexible TE device assembled of 4 Bi_2_Te_3_ legs (treated under the electric current of 0.5 A) was fabricated as schematically shown in the inset of Figure 5c. And Figure 5c shows the open circuit voltage and output power as a function of electric current at the temperature difference (*∆T*) ranging from 10 to 35 K. And a high temperature difference can be easily maintained between hot and cold side of thin films devices due to the polymide substrate (*κ* < 1 W m^−^^1^ K^−^^1^) [40]. As can be seen, the maximum open circuit voltage of ~7.96 mV can be achieved with the corresponding output power of 24.78 nW at *∆T* of 35 K. The performance of the flexible TE device can be evaluated by power density *P_density_* (*P_density_* = *P_max_*/w·h, where w and h represent the width and height, respectively) [40,41]. Figure 5d shows that the *P_density_* of the flexible TE device is 0.04 mW cm^−2^, 0.17 mW cm^−2^ and 0.36 mW cm^−2^, corresponding to the *∆T* of 10, 20 and 30 K, respectively.

## 4. Conclusions

In conclusion, we have successfully improved n-type Bi_2_Te_3_ f-TFs by P-ECT. It is found that, with the increase of electric current, the *n*_e_ increases and *σ* increases from 1874 to 2240 S cm^−1^. Consequently, the high *S*^2^*σ* of the Bi_2_Te_3_ f-TFs treated by 0.5 A achieves ~10.70 μW cm^−1^ K^−2^ at room temperature, which is competitive among the reported n-type Bi_2_Te_3_ f-TFs. Besides, a small *∆R*/*R*_0_ < 10% is achieved after bending test, suggesting high bending resistance of our prepared Bi_2_Te_3_ f-TFs. Subsequently, a flexible TE device composed of 4 n-type single legs generates an open circuit voltage of ~7.96 mV and an output power is 24.78 nW at Δ*T* of ~35 K. Our work demonstrates that P-ECT method can effectively further improve the electrical performance of Bi_2_Te_3_ f-TFs.

## Figures and Tables

**Figure 1 micromachines-13-01544-f001:**
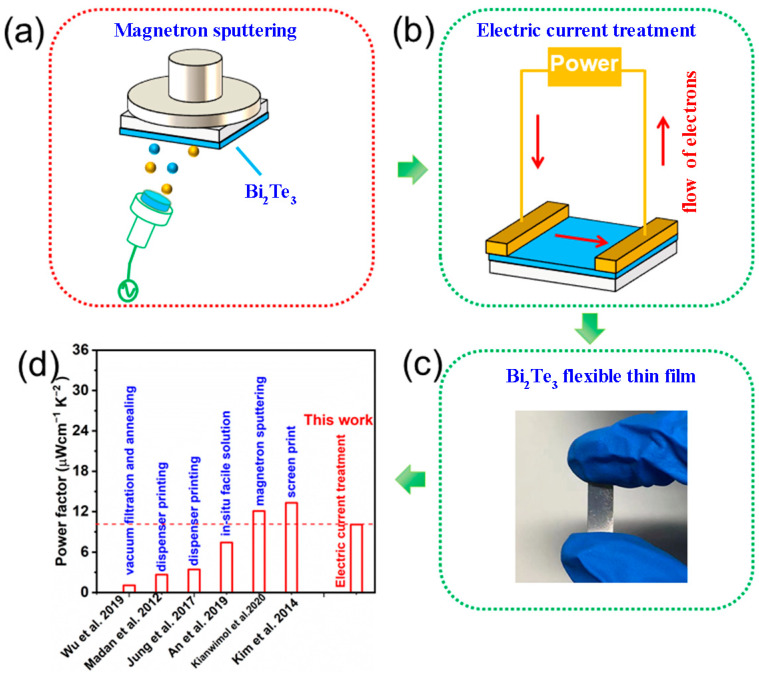
Schematic diagram of: (**a**) magnetron sputtering, (**b**) electric current treatment, (**c**) Bi_2_Te_3_ f-TF, (**d**) mechanically alloyed and dispenser printing (2012) [20]; screen print (2014) [32]; disperser printing (2017) [21]; vacuum filtration and annealing (2019) [18]; in situ solution (2019) [33]; magnetron sputtering (2020) [34].

**Figure 2 micromachines-13-01544-f002:**
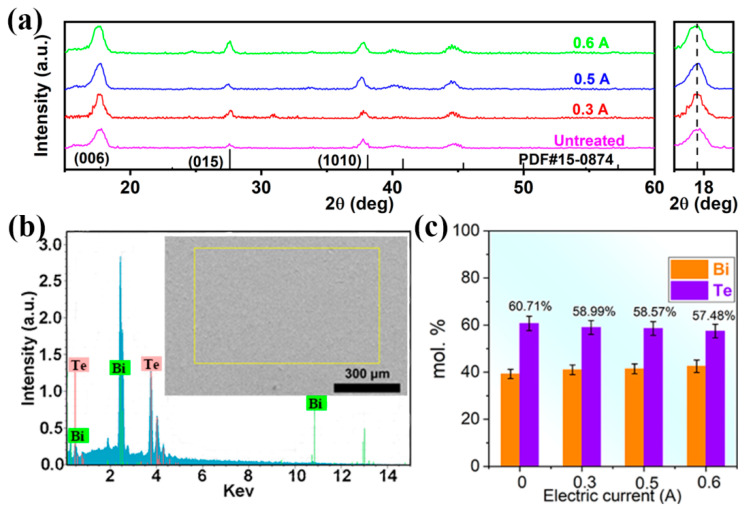
(**a**) XRD patterns of Bi_2_Te_3_ f-TF fabricated under different electrical intensities. (**b**) The SEM-EDS pattern. (**c**) The pattern of chemical stoichiometric ratio of Bi_2_Te_3_ f-TF.

**Figure 3 micromachines-13-01544-f003:**
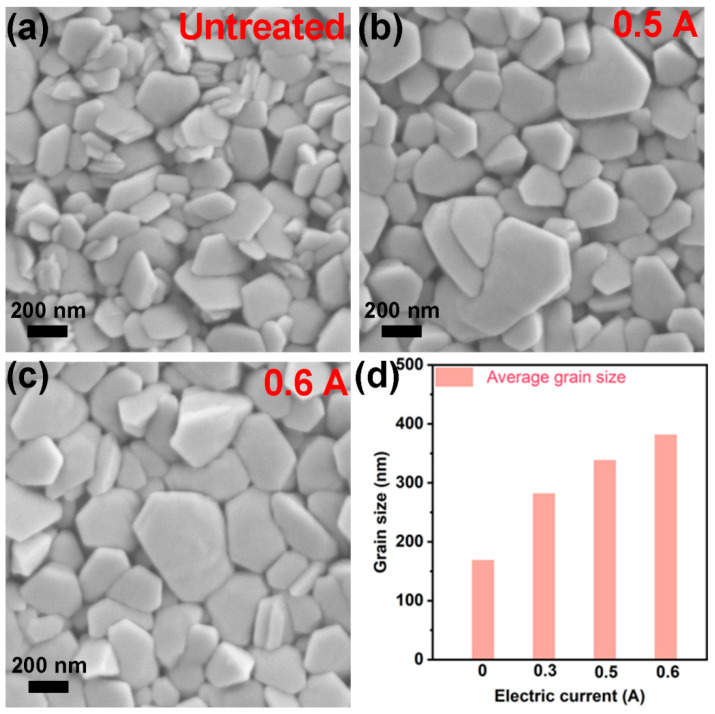
The SEM images of (**a**) untreated Bi_2_Te_3_ f-TF; (**b**) 0.5 A-Bi_2_Te_3_ f-TF; (**c**) 0.6 A-Bi_2_Te_3_ f-TF. (**d**) The average grain size of Bi_2_Te_3_ f-TF.

**Figure 4 micromachines-13-01544-f004:**
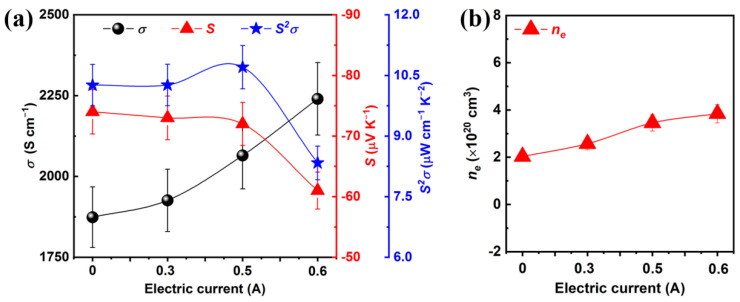
(**a**) The room temperature *σ*, *S*, and *S^2^σ* of Bi_2_Te_3_ f-TF as a function of electric current. (**b**) Room temperature *n_e_* of Bi_2_Te_3_ f-TF as a function of electric current.

**Figure 5 micromachines-13-01544-f005:**
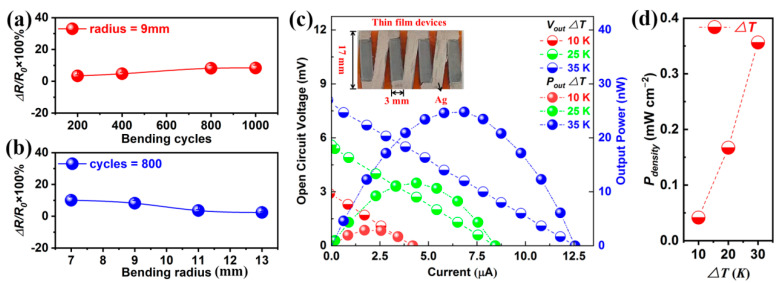
(**a**) The *∆R*/*R*_0_ as a function of bending cycles at bending radius of 9 mm. (**b**) The *∆R*/*R*_0_ as a function of bending radius at bending cycle of 800. (**c**) Open circuit voltage and output power as a function of electric current at different *∆T*. (**d**) The power density as a function of *∆T*.

**Table 1 micromachines-13-01544-t001:** SEM-EDS results of Bi_2_Te_3_ f-TF.

Electric Current	Thickness	Bi (at. %)	Te (at. %)
0 A	~580 nm ± 5 nm	39.29	60.71
0.3 A	~581 nm ± 5 nm	41.01	58.99
0.5 A	~583 nm ± 5 nm	41.43	58.57
0.6 A	~584 nm ± 5 nm	42.52	57.48

## Data Availability

The processed data required to reproduce these findings cannot be shared at this time as the data also forms part of an ongoing study.

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
