# Peer review of "Post-Electric Current Treatment Approaching High-Performance Flexible n-Type Bi2Te3 Thin Films"

_micromachines, 2022, doi:10.3390/mi13091544_

Round 1

Reviewer 1 Report

The manuscript submitted for the review process entitled Post-electric current treatment approaching high-performance flexible n-type Bi2Te3 thin films concerns the actual topic aimed at improving thermoelectric efficiency of flexible modules on the basis of classic TE material, namely bismuth telluride. The article is written thoughtfully and in a simple but still proper manner. The article offers a brief but sufficient introduction to the topic along with clearly stated objectives. The experimental part, as well as the discussion, are also written clearly and correctly. I have a few questions (listed below) and a few minor editorial mistakes (listed below) for the authors before the final publication of the article.

Factual questions:

1.     In the experimental part, the authors write about the thickness of the layers. How was the thickness of these layers determined (method, apparatus) ?

2.     The authors indicate that an increase of electric current during electric processing results in a decrease in Tellurium concentration. Although I agree with the mechanism of the formation of these defects, I have a question about whether the strong Te vacancy concentrations do not progress with successive measurement cycles, which may (perhaps there are literature data that confirm or negate this) ultimately lead to phase decomposition of the material and/or a decrease in the mechanical strength of the films?

3.     With reference to point 2, if further defecting of the material takes place, combined with increasing grain size, metallic Bi precipitates can be expected at grain boundaries. Maybe the authors can provide SEM photos of the microstructure after several measurement cycles? It would also be useful to measure the Seebeck coefficient and/or the carrier concentration of thin films after operation.

Minor editorial mistakes:

1.     Introduction line 52: 2.65 μWK-2 – the exponent “-2” is missing after “K”

2.     The quality of the lettering in Figures 1a, b and c is poor

I recommend the acceptance of this publication after a revision process (minor or major depending on the Editor's decision)

Author Response

We appreciate the referees’ comments and suggestions, and have tried our best to revise the manuscript. Please find our detailed replies to referees’ comments as shown in the word file.

Reviewer 2 Report

In this manuscript by Dongwei et.al., the authors report on a novel strategy to enhance thermoelectric performance in thin film Bi2Te3. The study is carried out well, with clear discussion. It is also intriguing that electric current treatment can enhance the electrical conductivity. I would recommend publication after the authors can address the minor points listed below:

- Does Joule heating arising from the electric current responsible for the "thermal annealing" effect that consequently improves the thin film crystallinity and therefore enhance conductivity?
This Joule heating effect needs to be ruled out.

- The Seebeck of thin film seems to be well below the usual value for bulk Bi2Te3. Why is this the case?

- In figure 3, the grain size growth seems to be consistent with thermal annealing effect. Is there any difference between thermal annealing and electrical current treatment?

- In figure 4, the magnitude of Seebeck decrease, while the magnitude of electrical conductivity increases. Such trade-off effects can be analyzed holistically by analyzing the weighted-mobility. The authors can refer to https://doi.org/10.1016/j.mtphys.2021.100519 for insights.

- In Figure 5, while the device performance is good, more discussion is needed regarding the effect of thin film thickness, and substrate effect. Authors can refer to https://doi.org/10.1016/j.mtener.2022.100964 for comparison. In the literature paper, the device performance is modelled to be dependent on film thickness. Is it the expected case in this work too?

Overall, this is a high quality work, and I would recommend acceptance after the above comments are addressed.

Author Response

(The authors gave the same response as above.)

Reviewer 3 Report

1. The main problem of manuscript is references include papers of Chinese researchers.

2. It is necessary tp analyze and include in references fresh fundamental manuscripts about high performance films on flexible substrate of Bi2Te3 based compounds of n- and p-type:

O. Kostyuk and et. and al. High thermoelectric performance of p-type BiTeSb films on flexible substrate. Materials Chemistry  and Physics. 253, 123427 (2021).

M. Maksaymuk and et. al. Development of the flexible film TE micro-TEG based on Bi2Te3 alloys. J. Materials Today. 21, 100753 (2021).

O. Kostyuk nd et. al. Development of thermal detector  based on flexible film TE module. J. Physics and Chemistry of Solid State. 22, 45 (2021).

3. Include in references papers of other researchers from these papers about Bi2Te3 films on flexible substrate.

Author Response

Response:We appreciate the positive comments to this work. The comments were extremely helpful to further improve our manuscript. 

Round 2

Reviewer 1 Report

The authors have responded to all my previous comments. Thank you for the clarification. The article can be successfully published in the present form.

Reviewer 3 Report

Presently it is OK